# Identification and Characterization of Variants in Intron 6 of the *LPL* Gene Locus among a Sample of the Kuwaiti Population

**DOI:** 10.3390/genes13040664

**Published:** 2022-04-09

**Authors:** Reem T. Al-Shammari, Ahmad E. Al-Serri, Sahar A. Barhoush, Suzanne A. Al-Bustan

**Affiliations:** 1Department of Biological Sciences, Faculty of Science, Kuwait University, Kuwait City 13060, Kuwait; reemt25@gmail.com (R.T.A.-S.); saharbarhoush@ku.edu.kw (S.A.B.); 2Kuwait Medical Genetic Center, Ministry of Health, Kuwait City 70051, Kuwait; 3Human Genetics Unit, Department of Pathology, Faculty of Medicine, Kuwait University, Kuwait City 46304, Kuwait; ahmad.alserri@ku.edu.kw

**Keywords:** LPL, sanger sequencing, CHD, CAD, BMI, population genetics, genetic diversity

## Abstract

Lipoprotein lipase (LPL) is responsible for the hydrolysis of lipoproteins; hence defective LPL is associated with metabolic disorders. Here, we identify certain intronic insertions and deletions (InDels) and single nucleotide polymorphisms (SNPs) in intron 6 of the *LPL* gene and investigate their associations with different phenotypic characteristics in a cohort of the general Kuwaiti population. Two specific regions of intron 6 of the *LPL* gene, which contain InDels, were amplified via Sanger sequencing in 729 subjects. Genotypic and allelic frequencies were estimated, and genetic modeling was used to investigate genetic associations of the identified variants with lipid profile, body mass index (BMI), and risk of coronary heart disease (CHD). A total of 16 variants were identified, including 2 InDels, 2 novel SNPs, and 12 known SNPs. The most common variants observed among the population were rs293, rs274, rs295, and rs294. The rs293 “A” insertion showed a significant positive correlation with elevated LDL levels, while rs295 was significantly associated with increased BMI. The rs274 and rs294 variants showed a protective effect of the minor allele with decreased CHD prevalence. These findings shed light on the possible role of *LPL* intronic variants on metabolic disorders.

## 1. Introduction

Lipoprotein lipase (LPL) is an important member of the lipase family, which plays a central role in hydrolyzing ester bonds of water-insoluble molecules, such as triglycerides (TG), cholesterol, and phospholipids. Lipids are insoluble in water and are therefore transported through the body’s circulation accompanied by proteins, hence the term “lipoproteins”. There are several classes of lipoproteins categorized based on their density, including chylomicrons, which are particles that consist of 99% triglycerides and thus have very low density [1]. Chylomicrons carry dietary lipids to peripheral tissues and the liver [1]. Other lipoproteins include chylomicron remnants, very low-density lipoproteins (VLDL), intermediate-density lipoproteins (IDL), low-density lipoproteins (LDL), and high-density lipoproteins (HDL). 

LPL is synthesized in the parenchymal cells of muscles, white and brown adipose tissues, heart skeletal muscle, lactating mammary glands, and macrophages [2] and is then relocated to the endothelium of nearby blood capillaries. The LPL enzyme catalyzes the breakdown of the TG-rich core of chylomicrons and VLDL into non-esterified fatty acids (NEFA) and monoglycerides. Fatty acids are absorbed by cells via cell surface receptors of the intestine, such as the cluster of differentiation 36/fatty acids translocase (CD36) [3]. Fatty acids are utilized for energy production in peripheral tissues, such as muscle, and are re-esterified and stored in adipose tissue. The breakdown of chylomicrons results in chylomicron remnants, which are broken down by the liver, while the breakdown of VLDL results in IDL. LPL is also important in facilitating the clearance of IDL and VLDL via LDL receptor-related protein in the liver (LDLR). LPL has also been shown to affect HDL metabolism [4,5].

It has been suggested that dysfunctional LPL could affect lipid homeostasis through a variety of metabolic and transport mechanisms [6], and its inactivity is associated with phenotypic abnormalities. Deficiency or low LPL activity influences the development of hypertriglyceridemia (HTG), which may lead to impaired endothelium-dependent vasodilatation, resulting in functional changes in myocardial perfusion [7]. In fact, a complete absence of LPL activity with a massive accumulation of chylomicrons in the plasma can result in familial LPL deficiency, a rare autosomal recessive disorder characterized by abdominal pain, acute pancreatitis, hepatosplenomegaly, and eruptive cutaneous xanthomata [8]. The severity of symptoms depends on the level of chylomicrons. In addition, homozygous LPL deficiency has been linked to severe HTG and marked reductions in high- and low-density cholesterol levels [9]. Xie and Li conducted a meta-analysis of 14 case-control studies investigating the *HindIII*, Ser447X, and *PvuII* polymorphisms of the *LPL* gene; the *HindIII* (rs320) polymorphism is an intronic base transition at position +495, which eradicates the *HindIII* restriction site [10]. The authors also found that the LPL *HindIII* polymorphism increased the risk of coronary artery disease (CAD) and that *HindIII* polymorphism also showed an association between plasma lipid levels and the risk of atherosclerosis, subsequently leading to coronary artery disease (CAD) and stroke [11]. The *PvuII* (rs285) LPL polymorphism, which is a C to T transition in intron 6 of the LPL gene, is associated with the risk of cardiovascular disease (CVD) [12].

The ~30 Kb *LPL* gene has been fully sequenced [13] after being mapped to chromosome 8p22 by in situ hybridization [14]. There are 10 exons in total, with exons 1–9 averaging between 105 and 276 bp, while exon 10 is 1948 bp. The *LPL* gene encodes the lipoprotein lipase enzyme, a non-covalent 55 kDa homodimer protein consisting of two structural domains: an amino-terminus domain and a carboxy-terminus domain [15]. The amino-terminus domain incorporates a lipid-binding site, ApoCII interaction site, a heparin-binding site, and a catalytic triad (a group of three amino acids, Ser132, Asp156, and His241) responsible for hydrolysis [16], while the carboxy-terminus domain also incorporates a heparin-binding region that is essential for attaching lipoproteins [17]. The protein consists of 448 amino acids in addition to a signal peptide that is 27 amino acids long [18]. 

Numerous mutations of LPL within the coding region, including missense, nonsense, and silent mutations, in addition to mutations in the non-coding regions (intronic splice donor variants and splice acceptor variants), have been identified and documented in different genetic databases. Mutations and polymorphisms within intronic regions usually generate alternative splice sites, resulting in abnormal protein synthesis. 

In this study, we focused on intron 6 of the *LPL* gene. Previous studies have reported a high frequency of short tandem repeat polymorphisms (STRs) in intron 6 of the LPL gene [19,20], which makes this region more prone to deletions or insertions. In addition, intron 6 is reported to be a recombination hotspot [21]. 

There are 70 intronic insertions and deletions (InDels) reported in intron 6 of the LPL gene according to the variant table found in Ensembl.org, 2018. Pirim et al. [22] re-sequenced the LPL gene in individuals with extreme HDL-C and TG levels and identified 176 mutations; the authors also found a total of 28 variants in intron 6 of the LPL gene, 6 of which were significantly (*p* < 0.05) associated with HDL-C and TG levels. Moreover, Pirim et al. [22] also reported that variants in intron 6 were InDels. Al-Bustan et al. [23] reported 293 variants by re-sequencing the LPL gene and found 252 single nucleotide polymorphisms (SNPs) and 39 InDels in the Kuwaiti population. Of the 293 variants, 37 were identified in intron 6 of the LPL gene, 8 of which were InDels.

This study identifies InDels and SNPs in intron 6 of the *LPL* gene in a large sample of the general Kuwaiti population. All identified variations were investigated for their potential association with lipid levels, BMI, and the risk of coronary heart disease (CHD) in the general Kuwaiti population.

## 2. Materials and Methods

### 2.1. Study Population

The population examined in this research consisted of 729 DNA samples from males and females from the general Kuwaiti population aged 18–87 years. For each sample, if applicable, clinical diagnoses, demographics, prescribed medications, and laboratory data, including lipid profiles, were obtained. Further categorization of the data was performed for association analysis by applying a defined inclusion criterion for 217 subjects with a confirmed diagnosis of CHD and 508 controls that were matched based on age and sex and had not been diagnosed with CHD at the time of sample collection.

The samples were collected from different hospitals in Kuwait between 2014 and 2017. Relevant phenotypic data were documented and registered in a database for each participant (Table 1). Each participant signed an informed consent form to participate in the study. The Local Ethical Committee at Kuwait University, along with the Ministry of Health Ethical Board, authorized this study. 

### 2.2. Epidemiological Analysis 

For each sample, gender, age, and BMI (kg/m²) were recorded using a standardized questionnaire administered to the participants. All the information was registered in a Microsoft Excel file and stored in the DNA bank database for future statistical analysis.

### 2.3. Sequencing Two Target Regions of Intron 6 at the LPL Gene Locus 

Two specific regions of intron 6 of the *LPL* gene were amplified using two pairs of primers that were originally designed for sequencing the full *LPL* gene [23]. For region 1, the forward primer was 5’-TGATTTGGGACTCGGGACCA-3’ and the reverse primer was 5’-CCTGTGTCGGTGTGTGTAAG-3’, amplifying the LPL gene from positions 8:19956194 to 8:19956691. For region 2, the forward primer was 5’-ACCGGAGGTTCTTGAGAAAA-3’ and the reverse primer was 5’-CGAACGAGGTCTACGATTCC3’, amplifying the *LPL* gene from position 8: 19958317 to 8:19958887. These pairs of primers amplified products of 498 bp and 571 bp, respectively. Appendix A provides the nucleotide sequences of the two regions.

The Applied Biosystems AmpliTaq Gold^®^ Master Mix (Applied Biosystems™, Thermo Fisher Scientific, Waltham, MA, USA) was used to amplify the DNA template using the GeneAmp™ PCR System 9700 (Applied Biosystems™, Thermo Fisher Scientific, Waltham, MA, USA). The PCR mixture was prepared as shown in Appendix A. The thermocycler conditions were based on the parameters shown in Appendix A. The BigDye Terminator v3.1 Cycle Sequencing Kit (Applied Biosystems™, Thermo Fisher Scientific, Waltham, MA, USA) was used for the sequencing reactions. The PCR mixture was prepared as shown in Appendix A. The thermocycler conditions were based on the parameters shown in Appendix A. The resulting PCR products were purified using the BigDye® XTerminator™ purification kit. Sanger sequencing was performed using a 16-capillary 3130xl Genetic Analyzer supported by ABI Sequencing Analysis Software v5.2. The resulting sequences were aligned to identify and genotype InDels and SNPs using the Clustal Omega Multiple Sequence Alignment tool in addition to the Ensembl genome browser to identify whether the identified InDels and SNPs were novel or reported in the annotated *LPL* gene. The functional role of the identified variants was also investigated using the online tools ClinVar, PheGenI, and Predict SNP2.

### 2.4. Statistical Analysis

#### 2.4.1. Determining the Genotype and Allele Frequencies

The genotypic and allelic frequencies of the detected InDels and SNPs of intron 6 of the LPL gene were calculated for the cohort (*n* = 729) using a simple gene counting method. The detected genotypes were tested for Hardy–Weinberg equilibrium (HWE) using an online-based Chi-square test calculator. 

#### 2.4.2. Testing the Association of the Detected InDels and SNPs with Phenotypic Variables

The potential associations of the identified variants and the phenotypic variables were statistically tested in the whole population (*n* = 729) in addition to the CHD (*n* = 217) subgroup along with their matched controls (*n* = 508). IBM Statistical Package for Social Sciences (SPSS) software v23.0 (IBM Corporation, Armonk, NY, USA) was used to investigate the association between the detected genotypes and other variables. The Kruskal–Wallis test was applied, and the results were reported as the mean  ± standard error. Furthermore, regression analysis models were used to check for possible associations between the detected variants and lipid profile parameters, BMI, and CHD risk while controlling for age and sex. The cut-off level for any statistical significance was a *p*-value of less than 0.05. 

## 3. Results

### 3.1. Identified InDels and SNPs

A total of 16 variants were identified, namely 2 InDels, 2 novel SNPs, and 12 known SNPs (Table 2). The most common variant was c.1019-685_1019-684insA (rs293), followed by c.1018+382_1018+383 insT or c.1018+382_1018+383 insA (rs274), c.1019-533A>C (rs295), and c.1019-646T>C (rs294) (Figure 1). SNPs with a minor allele frequency (MAF) of less than 0.05 were excluded from further analysis and from the genetic association, as they are less informative. The excluded SNPs were c.1019-c.1019-541G>A (rs901601579), c.1019-527G>A (rs296), c.1018+295C>A (rs138618627), c.1018+334C>G (rs272), c.1018+449T>G (rs540562340), c.1018+284A>C (rs144578061), c.1019-553A>G (rs910411725), c.1019-400T>C (rs297), c.1019-716G>A (rs292), and c.1019-827A>T (rs74746426), and two novel SNPs, one of which was found only in one sample (c.1018+397C>T, g.19956480C>T) and the other of which (c.1019-646T>G, g.19958614T>G) was observed in 10 samples (Figure 2).

### 3.2. Genotype and Allele Frequencies of the Selected LPL Variants

Four genetic variants of LPL, rs274, rs293, rs294, and rs295, were selected for statistical analysis with MAF ≥ 0.05. All deviated from HWE at *p* < 0.001, except for rs274 (*p* = 0.178) (Table 1). Further analysis was performed to investigate whether deviation was the outcome of a genetic association. The genotypic and allelic frequencies of the two InDels and the two SNPs were estimated in the study population (Table 3).

### 3.3. Genetic Association with Lipid Levels

Data cleaning included removing patients with missing lipid profiles and medical records of CHD, which reduced the sample size from 729 to 466. 

Analysis of the variants (Table 4) showed a significant difference in mean LDL with carriers of the minor allele of rs293 (−/A+A/A) (3.34 ± 0.86) in comparison to the major homozygous genotype (3.14 ± 0.88) in the dominant model at *p* = 0.024.

For further analysis of the significant genetic variant rs293 with lipid levels including LDL, linear regression was used, adjusting for age, sex, and BMI (Table 5), showing a significant positive correlation between carriers of the “A” insertion and elevated LDL levels (β = 0.191; 95% CI = 0.360–0.027) at *p* = 0.024.

### 3.4. Genetic Association with BMI 

The Kruskal–Wallis test was used to assess the difference in mean BMI between the three genotype groups of the four selected LPL variants in 725 samples (4 subjects were removed from the total cohort because of their missing BMI data) (Table 6), which showed a significant difference with both rs293 and rs295. Further analysis using linear regression showed that the recessive models of rs293 (*p* = 0.001) and rs295 (*p* = 0.007) were significantly associated with an increase in BMI by 4.032 and 3.318 units, respectively, adjusting for age and sex (Table 6).

### 3.5. Genetic Association with CHD 

Logistic regression was used to assess the association of the four selected variants with CHD under different genetic models in CHD cases (*n* = 217) vs. CHD-negative controls (*n* = 508). Table 7 shows a protective effect of the minor allele of rs293 in the dominant (*p* < 0.0001) and recessive (*p* = 0.013) models, which significantly decreased the prevalence of CHD by an OR of 0.009 and 0.078, respectively. The positive association with CHD was 18 times higher in carriers of the major (wildtype) homozygous genotype compared to other genotypes (*p* = 0.005). 

Similarly, carriers of the minor C alleles of rs294 were negatively associated with CHD at *p* = 0.004 and OR of 0.316 (0.146–0.686) (Table 8). Thus, carriers of the major (wildtype) homozygous genotype (TT) were three times more positively associated with CHD at the same *p*-value. 

The most informative genetic model assessing the association of rs274 was the dominant model (Table 9) with carriers of the major (wildtype) homozygous genotype significantly increasing the risk of CHD by five times at *p* < 0.0001, whereas carriers of either the A or T insertion were negatively associated with CHD with an OR of 0.179 (0.077–0.414) at *p* < 0.0001.

All odds ratios were adjusted for age and sex (female), which were significant predictor factors for CHD at *p* < 0.0001 for both, with ORs of 1.069 (1.055–1.083) and 0.357 (0.248–0.513), respectively.

## 4. Discussion

Generally, there has been less focus on intronic variants than on exonic variants, despite the frequent reports on their possible disease associations. Hence, this study provides insights into the important role of non-splice site intronic variants and their association with different phenotypic characteristics.

The MAF of the studied variants were compared to other previously reported frequencies. Comparing allele frequencies between populations is crucial since it provides insights into the demographic and evolutionary backgrounds of a population. In addition, comparing the rare (minor) allele frequencies can help to identify genomic regions influenced by natural selection, and verify the pathogenicity status of disease-causing variants. Table 10 lists the comparison of the minor allele frequencies between different populations of the detected variants in intron 6 of the *LPL* gene. The variants rs293 and rs295, though they appear to be in linkage disequilibrium, were analyzed independently and found to be associated with high LDL levels and high BMI in our studied population, respectively. Analyzing them independently allowed for the identification of their association with different metabolic disorders in our cohort. Moreover, both variants had a lower MAF (0.15, 0.12) in the Kuwaiti population compared to other populations, which suggests the possible independent pathogenicity of each variant and negative selection to eliminate their deleterious effects. The MAF of rs274 in the studied population is similar to that of African/African American populations while the MAF of rs294 is lower than most other populations. These differences in MAF between populations might be attributed to local adaptation to environmental selection pressures, which explains the disparities in health status and disease prevalence among different populations. 

Using the Human Splicing Finder tool to detect if the detected variants rs274, rs293, rs294, and rs295 reside in a potential splice site or any other splice sites accessories (enhancers or silencers), the results revealed no impact on splicing. However, other prediction algorithms were used, including ClinVar, PredictSNP2, Ensembl Variant Effect Predictor (VEP), and Phenotype-Genotype Integrator (PhenGenI), but no data were found except for the rs295 variant. PhenGenI is an online-based tool that merges the NHGRI genome-wide association study (GWAS) catalog data and other databases, including Gene, dbGaP, OMIM, eQTL, and dbSNP provided by the National Center for Biotechnology Information (NCBI). PhenGenI showed an association of rs295 with metabolic syndrome [24]; rs295 is also associated with both TG and HDL-C levels [22,24,25]. However, how these polymorphisms affect LPL function remains unknown, hence functional studies are warranted to assess the impact of the variants on *LPL* gene expression. It is difficult to determine the effects of these variants because the relationship between LPL activity and lipid profiles, BMI, and CHD risk is intricate.

The relationship between LPL and lipid levels is complex. LPL has been reported to be both protective and pathophysiological; LPL is considered protective due to its role in hydrolyzing circulating lipoproteins, which reduces TG and elevates HDL. Subjects with defective LPL showed an association with the progression of atherosclerosis [26]. In addition, LDL receptor or apoE knockout (KO) mice overexpressing LPL showed protection against atherosclerosis [27]. However, LPL also exhibits pathophysiological and pro-atherogenic actions, as proposed by Zilversmit in 1973 [28]. Zilversmit hypothesized that elevated local concentrations of cholesterol-rich remnants and LDL, resulting from the hydrolysis of TG-rich lipoproteins by LPL, would be taken into the arterial wall, thereby promoting the formation of atherosclerotic lesions due to the accumulation of cholesterol. Ichikawa et al. [29] created transgenic mice overexpressing human LPL to determine whether the increased activity of LPL increases susceptibility to atherosclerosis by observing lipid profiles and the formation of atherogenic lesions. The authors reported that these transgenic mice exhibited elevated levels of small, dense LDL particles (sdLDL); subsequently, the mice had greater aortic lesions than the control with normal LPL expression levels. LPL is highly active in macrophages in atherosclerotic lesions via transcriptional activation in addition to increased levels of apolipoprotein C-II (ApoCII), which enhances LPL function, leading to the accumulation of lipids [30,31,32]. 

The effects of the two commonly observed InDels (rs293 and rs274) and SNPs (rs294 and rs295) were investigated in our study population (*n* = 729) for their association with lipid profiles using regression analyses. Our results showed that the carriers of the rs293 minor allele (insertion of adenosine) were significantly associated with high LDL levels following a dominant genetic model. LPL plays a major role in the formation of LDL and hepatic lipases. LDL is a cholesterol-rich lipoprotein particle derived from VLDL and IDL, with a density ranging between 1.006 and 1.019 g/mL and size ranging between 25 and 35 nm. The variation in size and density of LDL particles has been previously investigated, and different associations with disease conditions have been linked to different LDL subclasses. sdLDL particles, a subclass of LDL, are pro-atherogenic and associated with obesity, HTG, low HDL, and metabolic syndrome [33]. sdLDL particles are metabolized over a period of five days, which is relatively slow. The slow clearance of sdLDL leads to its accumulation at the arterial walls by the function of macrophages, which leads to the formation of atherogenic plaques [34,35]. Since LPL plays a major role in lipoprotein metabolism, it impacts LDL particle size through its function in the catabolism of TG-rich lipoproteins, transforming them into IDL and LDL [36,37]. Hence, mutations in the LPL gene could disrupt this function and the formation of elevated levels of sdLDL particles [37]. 

Linear regression analysis showed that the recessive models of rs293 and rs295 were significantly associated with an increase in BMI by 4.032 and 3.318 units, respectively, after adjusting for age and sex. LPL plays a pivotal role in the regulation of body weight and composition through the enzymatic regulation of lipid partitioning. However, many conflicting results have been reported regarding the association between LPL activity, obesity, and body weight. The association between LPL activity and obesity might be tissue specific [38]. Jensen et al. created transgenic mice overexpressing LPL in skeletal muscles and observed a decrease in diet-induced weight gain through the prevention of lipid accumulation. However, higher levels of LPL expression resulted in severe myopathy and premature death of the transgenic mice due to a fatal accumulation of fatty acids [39]. Hara et al. [40] investigated the effects of the LPL activator NO-1886 on fructose-fed rats and found that TG levels were reduced without an increase in lipid accumulation. Furthermore, LPL activity and expression increased in very obese individuals who underwent weight loss, enhancing lipid accumulation and, hence, making further weight loss challenging [41]. Ferland et al. [42] found that adipose tissue LPL increased in response to a high-carbohydrate diet, which increased fat mass over 4 years in free-living adults.

In this study, the significant association of rs293 and rs295 was further examined for their effect on the development of metabolic disorders using CADD (https://cadd.gs.washington.edu/snv accessed on 27 March 2022). Both variants were benign based on the scores of 10.29 and 1.8, respectively. The cut-off value for pathogenicity is 20 and therefore neither variant would be considered causative alleles. In addition, we obtained the eQTL analysis from the genotype-expression project (https://gtexportal.org/home accessed on 29 March 2022) for the LPL rs295 which showed the C-allele (risk allele) to significantly (*p* < 0.00001) have a higher expression in adipose, skeletal, and epithelial tissues, which supports the results of this study.

Our study showed a protective effect of the minor alleles of rs293, rs294, and rs274, as they significantly decreased the risk of CHD (CAD). A positive association with CHD was higher in carriers of the major (wildtype) homozygous genotype. Functional validation of each variant is required to fully understand the effects of these variants on the risk of CHD. In the WHO Atlas of Heart Disease and Stroke, the burden of CHD from mortality and morbidity rose from 47 million in 1990 to 82 million in 2020. CHD is caused by the accumulation of cholesterol plaques inside the coronary arteries, which deliver blood to the heart muscle, thereby obstructing blood flow. Moreover, a sudden rupture of the plaques can occur, leading to the formation of blood clots. These clots could completely block blood flow to the heart, leading to heart attacks or strokes. Numerous candidate genes, including LPL, are associated with the risk of CHD [43]. 

It has been reported that in a population in southern Iran, LPL polymorphisms explain some lipid abnormalities found in CAD, including elevated TG levels and decreased HDL [44]. Seven LPL polymorphisms (T-93G, D9N, G188E, N291S, PvuII, HindIII, and S447X) were analyzed for their association with CHD via meta-analysis, which showed that some variants had opposing effects on HDL-C and TG levels, others had some beneficial profiles, and some were possibly associated with CHD [45]. Anderson et al. [46] conducted a study on CAD and myocardial infarction (MI) patients to examine the association between LPL polymorphisms and CAD and found a moderately strong association between the two. Patients who developed CAD in a seven-year follow-up study were examined for their serum LPL concentrations; the authors reported that these levels were inversely correlated with future CAD in seemingly healthy subjects [47]. Khera et al. [48] conducted a cross-sectional study on CAD case-control studies and identified a rare LPL mutation in 0.4% of the subjects, the presence of which was correlated with increased TG and occurrence of CAD. In another meta-analysis performed by Xie and Li [10], the S447 LPL variant was significantly associated with a high risk of CAD, while the PvuII polymorphism had no significant association.

## 5. Conclusions

Balanced lipid content in the human body is crucial for a healthy life. Lipids play an essential role in energy production and storage, hormone production, regulation of internal body temperature, and organ protection. Functional lipid metabolism is required to break down or manufacture lipids to maintain balance in the human body; hence dysfunction in the lipid metabolism process could lead to the development of different disease conditions. Therefore, when attempting to understand these disorders at the genetic level, researchers select candidate genes that are directly or indirectly involved in lipid metabolism. The LPL enzyme plays an important role in hydrolyzing ester bonds, similar to those found in lipids. Thus, it is a critical candidate gene for studying lipid metabolism and lipid profiles; therefore, disrupting mutations or polymorphisms in the LPL gene may cause serious dysfunction of the expressed protein and subsequent loss of proper lipid metabolism. 

Our study provides potential evidence of the role of intronic variants at the LPL gene locus in the fluctuation of lipid levels. The analysis of the four selected variants (rs274, rs293, rs294, and rs295) revealed an association with variations in lipid levels and CHD among the general Kuwaiti population with an 80% power for the sample size analyzed in the cohort as well as the case-control analysis based on a confidence interval set at 95% and a *p*-value of <0.05. The limitations of this study are mainly related to the lack of relevant data related to metabolic disorders. Further analysis on the identified variants, especially those with low MAF, remains warranted in larger cohorts. Our findings prompt efforts to improve the health of the Kuwaiti population, and, since drug efficacy and toxicity have a genetic component, these findings provide impetus to develop personalized preventive strategies and improve available treatments, or create new, more effective ones.

## Figures and Tables

**Figure 1 genes-13-00664-f001:**
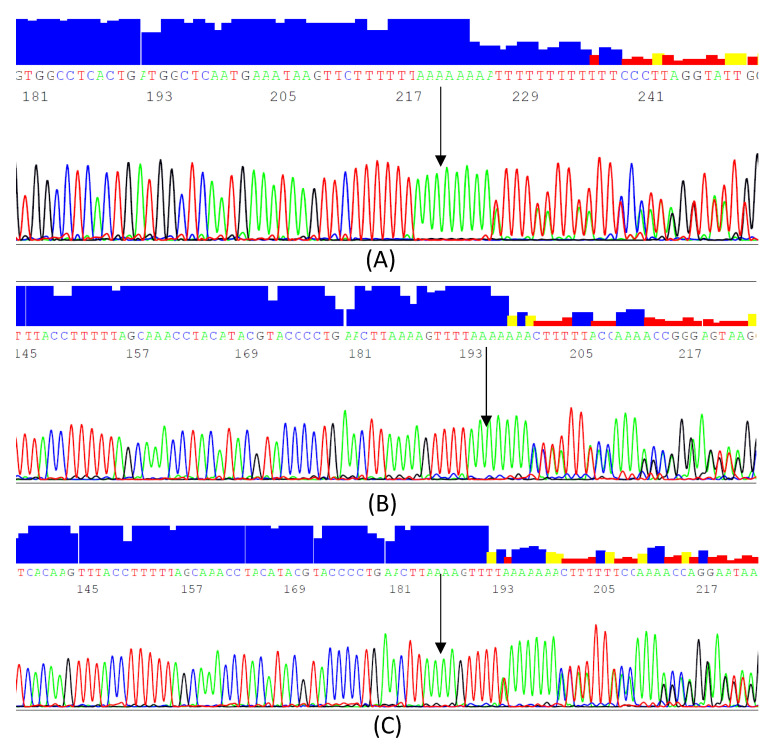
Sequencing electropherograms of the target regions (8:19956194-19956691 and 8:19958317-19958887) in intron 6 at the lipoprotein lipase (*LPL*) gene locus showing the InDels identified. (**A**) (rs293) c.1019-685_1019-684insA (forward strand); (**B**) (rs274) c.1018+386_1018+387insA (reverse strand); (**C**) (rs274) c.1018+386_1018+387insA (reverse strand). The vertical bars represent the Quality values (QV) designated to each base of the sequencing reaction analyzed by the software. The blue bars represent values of ≥20 (very good data), while the yellow and red bars represent values <20 (poor data).

**Figure 2 genes-13-00664-f002:**
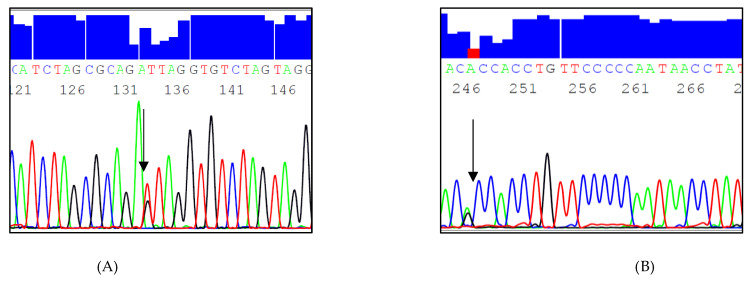
Sequencing electropherograms of the target regions (8:19956194-19956691 and 8:19958317-19958887) in intron 6 at the *LPL* gene locus showing known and novel single nucleotide polymorphisms (SNPs) (**A**) c.1019-533A>C (rs295) (reverse strand: T>G); (**B**) c.1019-646T>C (rs294) (reverse strand: A>G); (**C**) novel c.1019-646T>G, g.19958614T>G (forward strand); (**D**) novel c.1018+397C>T (reverse strand: G>A).

**Table 1 genes-13-00664-t001:** Clinical and epidemiological features of the study population (*n* = 729).

Subjects			
Gender	Males	Females	Total
Number	372	357	729
Age (Mean ± Std)	44.8 ± 16.1	44.2 ± 14.5	44.5 ± 15.3
**BMI *n* (%)**			
<25	80 (21.7)	50 (14)	130 (17.9)
25–30	135 (36.6)	96 (27)	231 (31.9)
>30	154 (41.7)	210 (59)	364 (50.2)

**Table 2 genes-13-00664-t002:** The sixteen identified variants in intron 6 (Chr8:19956194-19956691) of the *LPL* gene (NM_000237.3) with their location, type, predicted consequence, minor allele frequency (MAF) and global MAF, number of subjects with the mutated alleles among the study population (*n* = 729) and the calculated Hardy–Weinberg equilibrium (HWE). All genomic positions are derived from Ensembl GRCh38/hg38.

Variant	Location	Type	MAF	Global MAF *	Predicted Consequences	Number of Subjects	HWE
rs293	Chr8:19958568-19958575 c.1019-685_1019-684insA	Insertion A	0.15	0.25	Intronic variant	177	<0.001
rs274	Chr8:19956466-19956469 c.1018+386_1018+387insA	Insertion A	0.06	0.07	Intronic variant	93	0.178
rs294	Chr8:19958614 c.1019-646T>C	SNP (T>C)	0.07	0.13	Intronic variant	80	<0.001
rs295	Chr8:19958727 c.1019-533A>C	SNP (A>C)	0.12	0.27	Intronic variant	139	<0.001
rs144578061	Chr8:19956367 c.1018+284A>C	SNP (A>C)	0.03	<0.01	Intronic variant	37	0.483
rs74746426	Chr8:19958433 c.1019-827A>T	SNP (A>T)	0.02	0.02	Intronic variant	23	0.662
rs296	Chr8:19958733c.1019-527G>A	SNP (G>A)	<0.01	<0.01	Intronic variant	2	1
rs901601579	Chr8:19958719c.1019-541G>A	SNP (G>A)	<0.01	Not available	Intronic variant	1	1
rs138618627	Chr8:19956378c.1018+295C>A	SNP (C>A)	<0.01	<0.01	Intronic variant	1	1
rs272	Chr8:19956417c.1018+334C>G	SNP (C>G)	<0.01	0.02	Intronic variant	4	0.920
rs540562340	Chr8:19956532c.1018+449T>G	SNP (T>G)	<0.01	<0.01	Intronic variant	1	1
Novel rs294 genotype	Chr8:19958614 c.1019-646T>G	SNP (T>G)	0.01	Not available	Intronic variant	10	0.862
Novel SNP	Chr8:19956480 c.1018+397C>T	SNP (C>T)	<0.01	Not available	Intronic variant	1	1
rs292	Chr8:19958544 c.1019-716G>A	SNP (G>A)	<0.01	<0.01	Intronic variant	1	1
rs910411725	Chr8:19958707 c.1019-553A>G	SNP (A>G)	<0.01	<0.01	Intronic variant	1	1
rs297	Chr8:19958860c.1019-400T>C	SNP (T>C)	<0.01	0.25	Intronic variant	6	0.92

* Ensemble.org, 2018.

**Table 3 genes-13-00664-t003:** A summary of the observed genotypes, allele frequencies, *p*-value and HWE for the four variations, c.1019-685_1019-684insA (rs293), c.1018+382_1018+383insT or c.1018+386_1018+387insA (rs274), c.1019-533A>C (rs295), and c.1019-646T>C (rs294) in the whole population (*n* = 729).

Variable	Total *n* = 729 *n* (%)
rs274	
No insertion	636 (87.2)
Heterozygote insertion	81 (11.1)
Homozygote insertion	5 (0.7)
Allele frequencies wildtype/insertion	0.94/0.06
*p*-value HWE	0.178
rs293	
AA (wildtype)	552 (75.7)
Aa (Heterozygote mutant)	139 (19.1)
aa (Homozygote mutant)	38 (5.2)
Allele frequencies A/a	0.85/0.15
*p*-value HWE	<0.001
rs294	
TT (wildtype)	649 (89)
TC (Heterozygote mutant)	63 (8.6)
CC (Homozygote mutant)	17 (2.3)
Allele frequencies T/C	0.93/0.07
*p*-value HWE	<0.001
rs295	
AA (wildtype)	590 (80.9)
AC (Heterozygote mutant)	99 (13.6)
CC (Homozygote mutant)	40 (5.5)
Allele frequencies T/C	0.88/0.12
*p*-value HWE	<0.001

**Table 4 genes-13-00664-t004:** Comparison of means between the genotypes of the selected LPL variants lipid profiles and body mass index (BMI) in Kuwaiti Arabs (*n* = 466).

SNP	Variable	W/W	*n*	W/M + M/M	n	*p*-Value
rs274	TC	4.90 ± 1.02	389	4.92 ± 0.88	77	0.904
	TG	1.31 ± 0.98	389	1.24 ± 0.54	77	0.555
	HDL	1.06 ± 0.39	389	1.12 ± 0.42	77	0.213
	VLDL	0.55 ± 0.43	389	0.53 ± 0.24	77	0.626
	LDL	3.21 ± 0.90	389	3.22 ± 0.79	77	0.885
rs293	TC	4.85 ± 0.99	312	5.02 ± 1.01	154	0.092
	TG	1.35 ± 1.01	312	1.19 ± 0.70	154	0.069
	HDL	1.05 ± 0.37	312	1.11 ± 0.42	154	0.133
	VLDL	0.57 ± 0.44	312	0.50 ± 0.30	154	0.058
	LDL	3.14 ± 0.88	312	3.34 ± 0.86	154	0.024 *
rs294	TC	4.88 ± 0.96	406	5.10 ± 1.21	60	0.099
	TG	1.30 ± 0.94	406	1.26 ± 0.80	60	0.736
	HDL	1.06 ± 0.39	406	1.15 ± 0.39	60	0.090
	VLDL	0.55 ± 0.41	406	0.53 ± 0.34	60	0.693
	LDL	3.18 ± 0.86	406	3.38 ± 1.03	60	0.111
rs295	TC	4.87 ± 0.98	350	5.02 ± 1.04	116	0.156
	TG	1.32 ± 0.97	350	1.21 ± 0.75	116	0.257
	HDL	1.06 ± 0.39	350	1.11 ± 0.41	116	0.237
	VLDL	0.56 ± 0.43	350	0.51 ± 0.32	116	0.210
	LDL	3.16 ± 0.87	350	3.34 ± 0.89	116	0.059

* *p* < 0.05; -W/W indicates the major (wildtype) homozygous genotype; -W/M indicates the heterozygous genotype; -M/M indicates a minor (mutant) homozygous genotype.

**Table 5 genes-13-00664-t005:** A linear regression assessing the association between rs293 under a dominant genetic model with lipid levels adjusting for age, sex, and BMI in 466 Kuwaiti subjects.

Lipid	Variable	β-Coefficient	Lower 95% CI	Upper 95% CI	*p*-Value *
TC	rs293	0.167	−0.026	0.360	0.090
	Age	0.001	−0.005	0.007	0.697
	Sex (Female)	0.008	−0.177	0.192	0.936
	BMI	0.004	−0.007	0.016	0.464
TG	rs293	−0.078	−0.185	0.030	0.157
	Age	0.008	0.005	0.012	<0.0001
	Sex (Female)	−0.182	−0.284	−0.079	0.001
	BMI	0.014	0.007	0.020	<0.0001
HDL	rs293	0.044	−0.041	0.128	0.309
	Age	−0.001	−0.004	0.001	0.368
	Sex (Female)	0.232	0.152	0.313	<0.0001
	BMI	−0.004	−0.009	0.001	0.096
VLDL	rs293	−0.087	−0.197	0.023	0.122
	Age	0.008	0.005	0.012	<0.0001
	Sex (Female)	−0.171	−0.277	−0.066	0.001
	BMI	0.013	0.006	0.019	0.0001
LDL	rs293	0.191	0.021	0.360	0.027
	Age	−0.003	−0.008	0.003	0.308
	Sex (Female)	−0.145	−0.307	0.017	0.080

* *p* < 0.05.

**Table 6 genes-13-00664-t006:** A comparison of the mean BMI of the selected LPL genetic variants, followed by a linear regression assessing the association between rs293 and rs295 with BMI under a recessive genetic model adjusting for age and sex in 725 Kuwaiti Arabs.

SNP	W/W	*n*	W/M	*n*	M/M	*n*	Kruskal-Wallis *p*-Value	β- Coefficient (Recessive Model)	95% CI (Recessive Model)	*p*-Value (Recessive Model)
rs274	31.34 ± 7.63	633	33.18 ± 7.84	83	30.65 ± 7.98	9	0.091	_	_	_
rs293	31.31 ± 7.54	549	31.30 ± 7.23	138	35.80 ± 9.89	38	0.015 *	4.032	1.572 –6.491	0.001 *
rs294	31.53 ± 7.67	645	30.39 ± 6.28	63	36.27 ± 10.85	17	0.102	−	−	−
rs295	31.40 ± 7.55	587	30.89 ± 7.24	98	35.18 ± 9.52	40	0.024 *	3.318	0.906 –5.729	0.007 *

The recessive models for rs274 and rs294 were only represented by 9 and 17 samples, respectively. Therefore, they were not analyzed for significance. * *p* < 0.05.

**Table 7 genes-13-00664-t007:** Logistic regression analysis of the association of *LPL* rs293 with coronary heart disease (CHD) adjusting for age and sex in 725 Kuwaitis.

	Control (*n* = 508)	Case (*n* = 217)	OR (95% CI)	*p*-Value
Codominant Model				
−/−	65.5% (333)	99.5% (216)	18.125 (2.395–137.176)	0.005
−/A	27.2% (138)	0	<0.0001	0.996
A/A	7.3% (37)	0.5% (1)	1	
Dominant Model				
−/−	65.5% (333)	99.5% (216)	1	
−/A + A/A	34.5% (175)	0.5% (1)	0.009 (0.001–0.068)	<0.0001
Recessive Model				
−/− + −/A	92.7% (471)	99.5% (216)	1	
A/A	7.3% (37)	0.5% (1)	0.078 (0.010–0.591)	0.013

**Table 8 genes-13-00664-t008:** Logistic regression analysis of the association of *LPL* rs294 with CHD adjusting for age and sex in 725 Kuwaitis.

	Control (*n* = 508)	Case (*n* = 217)	OR (95% CI)	*p*-Value
Codominant Model				
TT	86.0% (437)	95.9% (208)	1	
TC	11.4% (58)	2.3% (5)	0.194 (0.072–0.526)	0.001
CC	2.6% (13)	1.8% (4)	1.025 (0.290–3.624)	0.970
Dominant Model				
TT	86.0% (437)	95.9% (208)	1	
TC+CC	14.0% (71)	4.1% (9)	0.316 (0.146–0.686)	0.004
Recessive Model				
TT+TC	97.4% (495)	98.2% (213)	1	
CC	2.6% (13)	1.8% (4)	1.115 (0.315–3.953)	0.866

**Table 9 genes-13-00664-t009:** Logistic regression analysis of the association of LPL rs274 with CHD adjusting for age and sex in 725 Kuwaiti Arabs.

	Control (*n* = 508)	Case (*n* = 217)	OR (95% CI)	*p*-Value
Dominant Model				
−/−	83.3% (423)	96.8% (210)	1	
−/A,T + A,T/A,T	16.7% (85)	3.2% (7)	0.179 (0.077–0.414)	<0.0001

**Table 10 genes-13-00664-t010:** Comparison of minor allele frequencies (MAF) of *LPL* rs274, rs293, rs294, and rs295 across different populations.

Variant	Kuwait	European (Non-Finnish)	European (Finnish)	African /African American	Latino/ Admixed American	East Asian	South Asian	Ashk-Enazi Jews
rs274	0.060	0.0004	N/A	0.0681	N/A	0.0000	N/A	0.0111
rs293	0.150	0.2614 *	N/A	0.3169 *	N/A	0.2341 *	0.208 *	N/A
rs294	0.070	0.1401	0.1313	0.1445	0.1580	0.1001	0.000	0.1840
rs295	0.120	0.2421	0.2301	0.3745	0.2298	0.2033	0.000	0.3586

* MAF were obtained from the 1000 genomes project (gnomAD database).

## Data Availability

The authors welcome any requests for additional data and documentation regarding the results of this study if not already provided in the supplementary files. Genetic data may be made available; however, no identifiers will be provided as per patient rights for confidentiality.

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
