# Peer review of "Identification and Characterization of Variants in Intron 6 of the LPL Gene Locus among a Sample of the Kuwaiti Population"

_genes, 2022, doi:10.3390/genes13040664_

Round 1

Reviewer 1 Report

This paper aims to characterize correlations between intronic frequent variants of LPL gene and various clinical or biological parameters. It is firmly argumented and explained. It helps to understand how deep intronic variations, rarely cared in molecular diagnostics routine, can influence phenotype.

Some minor points remain unclear or can be improved.

  • L73. Mention of Figure 1.3 is false
  • Table 2. Authors should use the international HGVS nomenclature. Variant nomenclature is incomplete or unclear. When genomic coordinates are used, reference version of the genome must appear, as well as number of chromosome. When coding coordinates are used, transcript number must be given... Please use the site https://varnomen.hgvs.org/
  • Table 2. Since most variants are frequent, it could be interested to compare frequencies in Kuwaity and general population. gnomad browser provides such information.
  • Table 3.
    • Titles lines may be highlighted or put in a different format to improve lisibility
    • A legend should explain to which genotypes do AA, Aa and aa correspond, since it seems unclear to me
  • All results should be given in the same order, by variant number for instance (or by frequency, whatever...). It is uncomfortable to read a result for variant rs293 first, then for rs294 first... one have to reorganize all the informations.
  • L292. There is a confusion in the variant numbers.
  • L330. The insertion concerns an Adenosine, not the base Adenine.
  • Table 5 and followings : please, explain what exactly means "adjusted for age, sex and BMI" ? The age ranges from 18 to 87, so it seems that adjusting for age could have an impact... Remains unclear to me.

Author Response

First we would like to express our appreciation for the time and valuable comments provided by the reviewers and for their positive feedback. Below is a point by point response the reviewer's comments.  All the changes made in the  manuscript are marked by track review in the revised version.

  • L73. Mention of Figure 1.3 is false
    • This has been deleted.
  • Table 2. Authors should use the international HGVS nomenclature. Variant nomenclature is incomplete or unclear. When genomic coordinates are used, reference version of the genome must appear, as well as number of chromosome. When coding coordinates are used, transcript number must be given... Please use the site https://varnomen.hgvs.org/
    • The following information were added to the table in the revised manuscript: NM accession number (NM_000237.3), coordinates of the amplified region with chromosome number included (Chr8:19956194-19956691), the reference version of the genome (GRCh38/hg38), and the chromosomal location for each variant.

  • Table 2. Since most variants are frequent, it could be interested to compare frequencies in Kuwaity and general population. gnomad browser provides such information.
    • We agree with this suggestion and thank the reviewer for pointing it out. A table (Table 10) and statements regarding this and how the differences in allele frequencies may be influenced by evolutionary processes has been added to the discussion.
  • Table 3.
    • Titles lines may be highlighted or put in a different format to improve visibility
      • Done
    • A legend should explain to which genotypes do AA, Aa and aa correspond, since it seems unclear to me
      • The following classifications were added to clarify the genotypes: wild-type, heterozygote mutant, and homozygote mutant

  • All results should be given in the same order, by variant number for instance (or by frequency, whatever...). It is uncomfortable to read a result for variant rs293 first, then for rs294 first... one have to reorganize all the informations.
    • Corrected. All the results follow the same order: rs274, rs293, rs294, rs295 in the revised manuscript.

  • L292. There is a confusion in the variant numbers.
    • Corrected
  • L330. The insertion concerns an Adenosine, not the base Adenine.
    • Corrected
  • Table 5 and followings : please, explain what exactly means "adjusted for age, sex and BMI" ? The age ranges from 18 to 87, so it seems that adjusting for age could have an impact... Remains unclear to me.
    • We have controlled/adjusted for common variables that are known to increase the risk of such phenotypes. As shown in our own cohort analysis that age, BMI and gender have all been found significantly to be a predictor variable that increases the risk of dyslipidemia. We controlled for these variables along with our selected variants within the regression model to show that our variant is significant even when all these known variables (age, BMI and gender) are included in the model, supporting that the variant as a predictor factor.

Reviewer 2 Report

The manuscript by Al-Shammari and colleagues reports on the genetic association analysis of four selected variants (rs274, 404 rs293, rs294, and rs295) identified by sequencing of a region of intron 6 of the LPL gene in individuals of the Kuwaiti population, with lipid levels and CHD. The authors found association evidence for one variant, rs293, with LDL and TG, HDL and VLDL; for rs293 and rs295 with BMI; for rs293, rs294 and rs274 with CHD. The main problem of this study, as stated by the authors in the discussion, is sample size. In addition, the novelty of the reported findings is limited and lastly there is no functional follow up  of the reported variants.

Here is a list of points that require further clarification by the authors.

  1. Please, provide power calculations for the two datasets used (general population and case-control cohorts), for the detection of a positive association at p=0.05 for the frequency of the 4 variants selected for genetic follow-up.
  2. Please, indicate the position of the variants based on the last genome release (GRCh38/hg38). For example, for SNP rs295: chr8:19958727-19958727.
  3. Please, check and indicate the LD for the 4 variants (rs293, rs294, rs295 and rs274) selected for further genetic analysis. For instance, the two SNP variants rs293 and rs295 are in complete LD (r2=1 in Asians, see LinDA browser) and are therefore perfect proxies. Their genetic effects on the tested phenotypes are indistinguishable and the lack of nominal significance in some of the analyses (i.e., association with lipid levels) is very likely due to low sample size in all analyses performed. For instance, the rs295/293 SNPs have been previously associated by GWAS with TG and other lipids (Harshfield et al., 2021) as well as with metabolic syndrome (Kraja et al., 2011) in large cohorts. Please, state.
  4. In the absence of functional studies, please report CADD score of the variants and check for their effect as eQTLs for the expression of the LPL gene in different tissues. This seems to be the case for variants rs295/rs293.

Author Response

We would like to express our appreciation for the time and valuable comments provided by the reviewers and for their positive feedback. Below is a point by point response the reviewer's comments.  Kindly note that the manuscript has been revised and edited for language by Editage. All the changes made in the  manuscript are marked by track review in the revised version. 

  1. Please, provide power calculations for the two datasets used (general population and case-control cohorts), for the detection of a positive association at p=0.05 for the frequency of the 4 variants selected for genetic follow-up.
  • Using Epi Info https://www.openepi.com/Menu/OE_Menu.htm we have calculated the sample size for the general population and case-control cohort.

    For the general population: After taking the average mean and standard deviation of the associated variants with the phenotypes for a confidence interval set at 95% and a p-value set at < 0.05. The power study calculation showed in order to reach a power of 80 percent then an average of sample size of 638 is required. The cohort in this study included a total of 725.

    For the case-control cohort: After setting the case-control ratio, power and p-value along with providing the number of exposed samples to the risk allele. The power calculation detected that we needed a maximum of 140 cases and 321 controls do provide 80% power. The number of cases in this study was 217 and 508 for the controls.
  • A statement regarding the power of the sample size has been added in the discussion of the revised manuscript.
  1. Please, indicate the position of the variants based on the last genome release (GRCh38/hg38). For example, for SNP rs295: chr8:19958727-19958727.
  • The variants positions have been modified for all the variants in the revised manuscript.
  1. Please, check and indicate the LD for the 4 variants (rs293, rs294, rs295 and rs274) selected for further genetic analysis. For instance, the two SNP variants rs293 and rs295 are in complete LD (r2=1 in Asians, see LinDA browser) and are therefore perfect proxies. Their genetic effects on the tested phenotypes are indistinguishable and the lack of nominal significance in some of the analyses (i.e., association with lipid levels) is very likely due to low sample size in all analyses performed. For instance, the rs295/293 SNPs have been previously associated by GWAS with TG and other lipids (Harshfield et al., 2021) as well as with metabolic syndrome (Kraja et al., 2011) in large cohorts. Please, state.
  • The authors agree that similar to the two SNPs rs293 and rs295 being in LD in Asians and Europeans, that the data in our study also appears to be in LD. However, the other SNPs rs274 and rs294 are not in LD with any of our selected studied SNPs. Based on that we decided to perform the statistical analysis for each SNP independently with the different metabolic disorders which allowed the identification of the association of the reported SNP's with different metabolic disorders in the cohort. A statement regarding this point has been added to the discussion of the revised manuscript.  
  1. In the absence of functional studies, please report CADD score of the variants and check for their effect as eQTLs for the expression of the LPL gene in different tissues. This seems to be the case for variants rs295/rs293.
  • The authors opted not to perform this analysis as the study is reporting each variant as an independent genetic factor.

Round 2

Reviewer 1 Report

I would like to thank the authors for their positive implementation of the manuscript. Tables are now much clear to read, genomic coordinates are properly used and MAF of variants are interestingly analyzed by ethnicity.

Author Response

The author's appreciate the reviewer's comments and positive feedback.

No changes are necessary.

Reviewer 2 Report

In the revised version of the manuscript number  "genes-1620950" , Dr. Al-Shammari and colleagues have addressed most of the points I raised and I have no further comments. I just recommend that they explain in more details how  the eQTL analysis of the variant rs295 was performed,  as it is not completely clear from the paragraph added in the new version.

Author Response

The author thank the reviewer for the positive feedback and apologize for the confusion. The eQTL for rs295 was  done using an online tool (Genotype tissue Expression Project). Accordingly, minor changes have been made to clarify this point in the newly revised manuscript as highlighted by track review.

Also note that the typo's have been corrected and the manuscript was again sent to Editage for language editing of the new revised manuscript.